# Urinary Exosomal MicroRNAs as Biomarkers for Obesity-Associated Chronic Kidney Disease

**DOI:** 10.3390/jcm11185271

**Published:** 2022-09-07

**Authors:** Angel Earle, Madison Bessonny, Josh Benito, Kun Huang, Hannah Parker, Emily Tyler, Brittany Crawford, Nabeeha Khan, Bridget Armstrong, Alexis Stamatikos, Sudha Garimella, Alyssa Clay-Gilmour

**Affiliations:** 1Department of Epidemiology & Biostatistics, Arnold School of Public Health, University of South Carolina, Columbia, SC 29208, USA; 2Prisma Health, Pediatric Nephrology, Greenville, SC 29615, USA; 3Department of Food, Nutrition, and Packaging Sciences, College of Agriculture, Forestry & Life Sciences, Clemson University, Clemson, SC 29634, USA; 4Department of Exercise Science, Arnold School of Public Health, University of South Carolina, Columbia, SC 29208, USA

**Keywords:** exosomes, urinary, microRNA, exosomes, biomarkers, obesity

## Abstract

The early detection of chronic kidney disease (CKD) is key to reducing the burden of disease and rising costs of care. This need has spurred interest in finding new biomarkers for CKD. Ideal bi-omarkers for CKD should be: easy to measure; stable; reliably detected, even when interfering substances are present; site-specific based on the type of injury (tubules vs. glomeruli); and its changes in concentration should correlate with disease risk or outcome. Currently, no single can-didate biomarker fulfills these criteria effectively, and the mechanisms underlying kidney fibrosis are not fully understood; however, there is growing evidence in support of microRNA-mediated pro-cesses. Specifically, urinary exosomal microRNAs may serve as biomarkers for kidney fibrosis. In-creasing incidences of obesity and the recognition of obesity-associated CKD have increased interest in the interplay of obesity and CKD. In this review, we provide: (1) an overview of the current scope of CKD biomarkers within obese individuals to elucidate the genetic pathways unique to obesi-ty-related CKD; (2) a review of microRNA expression in obese individuals with kidney fibrosis in the presence of comorbidities, such as diabetes mellitus and hypertension; (3) a review of thera-peutic processes, such as diet and exercise, that may influence miR-expression in obesity-associated CKD; (4) a review of the technical aspects of urinary exosome isolation; and (5) future areas of research.

## 1. Introduction

It is currently estimated that 37 million people have chronic kidney disease (CKD), and 48% of those with severely reduced kidney function are unaware [1]. CKD is missed early in the course of the disease for multiple reasons. First, the disease process is indolent, causing patients to seek help late in the process when irreversible harm has already taken place. Second, the diagnostic tests used in current clinical practice are fraught with pitfalls, causing delays in recognition and treatment. The most common diagnostic measure is serum creatinine, which is often used as a surrogate for the estimated glomerular filtration rate (eGFR) for CKD staging and progression [2]. This test must be interpreted differently for different age groups and body habitus, and the observed values can vary with collection time/hydration status [3]. Other diagnostic criteria, such as urinary protein and microalbumin levels, can be attributed to causes other than CKD, and require an accurate first morning sample or 24 h urine collections [4]. Often, by the time these tests are noted to be abnormal, CKD is established [4], as underscored by 2017 data reporting that treating Medicare beneficiaries with CKD cost over USD 84 billion [1].

The early detection of CKD is key to reducing the burden of disease and rising costs of care. This identified need has spurred interest in finding new biomarkers for CKD. Furthermore, the increasing incidence of obesity and the recognition of obesity-associated kidney disease has increased interest in the interplay of obesity and CKD [5]. In this review we provide: (1) an overview of the current scope of CKD biomarkers within obese individuals to elucidate the genetic pathways unique to obesity-related CKD; (2) a review of microRNA expression in obese individuals with kidney fibrosis in the presence of comorbidities such as diabetes mellitus and hypertension; (3) a review of therapeutic processes such as diet and exercise that may influence miR-expression in obesity-associated CKD; (4) a review of the technical aspects of urinary exosome isolation; and (5) future areas of research.

### 1.1. Current Scope of CKD Biomarker Research

An ideal biomarker for CKD should be: easy to measure; stable; reliably detected even when interfering substances are present; site-specific based on the type of injury (tubules vs. glomeruli); and its changes in concentration should correlate with disease risk or outcome. Currently, there is no single candidate that fulfills these criteria effectively [6].

Approaches to finding valid biomarkers have centered around two main philosophies. Unbiased approaches involve harnessing the power of proteomics and metabolomics to help identify candidate biomarkers that are changed by CKD. In contrast, biased approaches are based on hypotheses related to glomerular or tubular injury pathophysiology [7]. Plasma biomarkers of tubular injury (KIM-1), repair (YKL-40), or inflammation (MCP-1, suPAR, TNF receptor-1 (TNFR-1), and TNFR-2) may identify children with CKD at risk for glomerular filtration rate (GFR) decline. Urinary biomarkers, including α1-microglobulin, KIM-1, and TFF-3, have been found to have the most clinical correlation with mortality in adult CKD patients [6]. Goknar et al. evaluated early urine kidney injury markers (microalbuminuria, NAG, NGAL, and KIM-1) in obese children and reported that obese children had higher urinary NAG and KIM-1 levels compared with lean controls [8]. The quest to detect specific unique kidney biomarkers in obese children is more difficult, because they are more likely to have an overlap with cardiac or metabolic factors as opposed to lean children [8].

### 1.2. MicroRNAs as Candidate Biomarkers for Kidney Fibrosis in Obesity

Kidney fibrosis is the buildup of scar tissue within the parenchyma, and it represents the common final pathway before end-stage kidney disease (ESKD). Indeed, cortical interstitial expansion is the best histologic predictor of kidney functional decline in CKD [9]. The mechanism underlying kidney fibrosis is not well understood; however, there is growing evidence in support of a microRNA-mediated process [10]. Specific microRNAs are involved at different stages of kidney fibrosis via different biological pathways [9]. MicroRNA expression profiles may be utilized in clinic as biomarkers in body fluid (blood/urine) for the early detection of kidney fibrosis and, thus, CKD. The identification of noninvasive biomarkers for the early diagnosis of kidney disease, such as urinary microRNA, would be clinically significant, specifically for feasibility in pediatric populations. Several of the urinary exosomal microRNAs that may serve as potential biomarkers for kidney fibrosis (CKD) and obesity are described herein (miR-21, miR-29, miR-146, and miR-200). For our review, we used the HMDD (the Human microRNA Disease Database) v3.2 (http://www.cuilab.cn/hmdd, accessed on 23 August 2022) [11], which curates experiment-supported evidence for human microRNA (miRNA) and disease associations. We searched for miR associated with CKD and obesity in the HMDD. Furthermore, we performed a comprehensive literature review based on references from the HMDD using PubMed, including the search terms “(chronic kidney disease OR CKD) AND (obesity) AND (microRNA OR miR OR miRNA)” from 2010 onwards, written in English-language journals. Our goal was to review miR which had an overlap of published associations with both CKD and obesity. Figure 1 shows the overlap of candidate microRNAs and pathways associated with obesity, inflammation, and kidney-specific outcomes.

### 1.3. miR-21

miR-21 is highly expressed in kidney tissue and is upregulated in the urine/blood of patients with a degree of kidney fibrosis [9,10,12,13]. MiR-21 has also been used as a marker for kidney fibrosis in kidney transplantation recipients [9,14,15], and has been observed to be upregulated in patients who are undergoing peritoneal dialysis (treatment for those experiencing kidney failure) [16]. Within the miR-21 family, miR-21-5p has been found to be associated with fibrosis and kidney survival in patients with IgA nephropathy (an autoimmune disease that targets kidneys) [17]. MiR-21 expression is closely linked to transforming growth factor (TGF)-β1 signaling, a key pathway for kidney fibrosis, in which TGF-β1 induces the upregulation of miR-21 [9,10,12,16,18,19]. The TGF-β/Smad3 signaling pathway is also implicated in obesity, inflammation, and diabetes [20]. The TGF-β/Smad3 mechanistically links many comorbidities with obesity through its profibrotic, remodeling, and proinflammatory functions [21]. MiR-21 is associated with obesity-related inflammation (Figure 1A) [22]; miR-21 inhibition has been shown to be associated with reduced body weight, as well as adipocyte size and serum triglycerides [23]; miR-21 is upregulated in subcutaneous adipose tissue in studies of human obesity [24].

### 1.4. miR-29

Members of the miR-29 family have been found to be predictive urinary markers for kidney fibrosis and kidney-specific inflammatory states. Urinary miR-29b is significantly downregulated in IgA nephropathy patients compared to healthy controls [15]. Urinary miR-29c downregulation was associated with early kidney fibrosis in patients with lupus nephritis (LN) and was correlated with both kidney function and the degree of fibrosis in CKD patients [25,26]. MiR-29c and miR-29b impact kidney fibrosis via the TGF-β/Smad3 signaling pathway, as demonstrated in a number of in vivo and in vitro models [27,28]. The suppression of miR-29 expression by TGF-β and SP1 promotes collagen production and kidney fibrosis [27,28,29]. Simultaneously, the miR-29 family has also been observed in obesity-related inflammation pathways (Figure 1A) [22,30]. A miR-29c study identified the involvement of miR-29c as a marker for maternal obesity and suggested a possible role of such for pre- and postnatal growth, through similar obesity–inflammation pathways [30].

### 1.5. miR-200

Inconsistencies have been found regarding the role of miR-200 in kidney fibrosis in mouse model studies [9,10]. MiR-200 appears to play a role in glomeruli epithelial mesenchymal transformation (EMT) via the TGF-B signaling pathway, an important aspect of regular kidney function [7,9]. In humans, a 2017 study by Zununi et al. found that the dysregulation of urinary miR-200b (and miR-21) was associated with interstitial fibrosis and tubular atrophy in kidney transplant recipients [31]. MiR-200 has also been found to be an important target for obesity, in that it is linked to the alteration of leptin and insulin signaling, specific to the upregulation of hypothalamic miR-200a [32]. As described previously, TGFB-Beta/Smad3 signaling has been implicated in various obesity–inflammation pathways, including the endocrine organ secretion of adipocytokines (e.g., leptin and resistin) (Figure 1A) [33].

Given the overlapping microRNAs implicated in kidney fibrosis and obesity-related pathways (Figure 1A); future research should focus on delineating the relationship and potential clinical use of urinary microRNAs as biomarkers for high-risk individuals for kidney fibrosis and future CKD.

### 1.6. miR-146

miR-146 is a known inflammation-related microRNA (Figure 1B) [6,9]. MiR-146a’s upregulated expression has been observed in kidney and urinary samples from diabetic nephropathy (DN)/glomerular endothelial injury; IgA nephropathy patients; lupus nephritis; and kidney transplant recipients [9,34,35,36]. MiR-146a is regulated via the inflammatory pathway NF-kappa B. NF-κB is a key mediator of the inflammatory response, and its dysregulation has been associated with kidney fibrosis [37,38]. Under proinflammatory conditions, miR-146a is transcriptionally upregulated via Toll-like receptor IL-1 through the activation of the NF-kappa B [9,37,38]. Similarly, as observed in kidney-specific inflammation, the dysregulation of miR-146 may contribute mechanistically to the heightened inflammatory state associated with overweight and obesity [39].

## 2. MicroRNA Expression in Obese Individuals with CKD in the Presence of Comorbidities

The expression of CKD microRNAs in an obese patient is complex, as it is not uncommon for obese patients to have comorbidities and lifestyle factors that may impact the CKD etiological pathway. Given the well-documented biological associations amongst obesity, diabetes mellitus (especially as it affects nephropathic pathways), and impaired vascular function, genetic markers associated with these conditions may also have a role to play in identifying CKD in obese individuals. The families of mir-126 and miR-770 have been shown to robustly predict the progression of diabetic nephropathy [40]. Individually, miR-126 is typically expressed in endothelial cells and has been shown to be significantly associated with vascular-related processes and diseases [41,42]. In relation to CKD, miR-126 has been shown to have altered levels in the serum and in the brain of mice with CKD [42]. Studies with human participants have supported this association, showing that plasma levels of miR-126 (more specifically, miR-126-3p) were significantly higher in those with diabetic kidney disease (DKD) compared to those without it [43]. The pathway for miR-126 is also related to miR-223 expression. miR-223 is a hematopoietic factor regulating the cardiac glucose metabolism and cholesterol homeostasis [44]. After accounting for estimated GFR levels, lower levels of miR-126 or miR-223 have been associated with lower survival rates in patients with CKD [42].

Additional examples of the multifaceted relationships between CKD, obesity, and comorbidities include the miR-let-7b and miR-148b pathways and miR-103a-3p. Initial experimental evidence has shown miR-148b and miR-let-7b to regulate the O-glycosylation process of immunoglobulin A1 in patients with IgA nephropathy [45], while, later, cohort-based research demonstrated that renal miR-148b levels were significantly and independently correlated with CKD progression in IgAN patients [46]. miR-103a-3p has been extensively researched and connected with a myriad of regulatory cellular processes, such as cellular stress, angiogenesis, and cell division, as well as being significantly associated with diabetes, cancer, and Alzheimer’s disease [47]. Within the CKD context, miR-103a-3p has been shown to play a significant role in the development of hypertensive nephropathy [48], leading to the discovery of a novel miR-103a-3p/SNRK pathway involving angiotensin II, contributing to kidney injury [49].

## 3. Effect of Diet and Exercise on MicroRNA Expression

While comorbidities may confound the biological pathways underlying obesity and CKD, ameliorative health factors may also do the same. Just as urinary exosomal microRNAs may serve as biomarkers for kidney fibrosis, the expression of microRNA may also be influenced by diet and activity levels impacting obesity. Diet can alter protein synthesis and affect the proteins needed for microRNA processing, and there is evidence to suggest changes in diet can alter metabolism and, thus, change microRNA expression. Given that a change in diet and exercise routine may be commonly recommended to obese patients with CKD, robust screening panels for this patient population must take these factors into consideration.

Several key enzymes along the pathway may also be altered by diet [50]. Changing dietary factors may lead to favorable changes in the microRNAs involved in kidney fibrosis, even in the absence of demonstrable weight loss. Thus, microRNAs may represent independent biomarkers for kidney health not influenced by adiposity. Various diets and dietary interventions, including high-fat (Western) diets, caloric restriction (CR), and the use of bioactive micronutrients and plant derivatives, have been associated with epigenetic changes that alter cellular signaling. These include DNA methylation, histone acetylation, and changes in microRNA expression [51].

Families of microRNA known to be altered with specific diets include the miR-17, miR-21, and miR-200 groups. The miR-200 family, which includes miR-200b, miR200c, miR-429, and miR-548a, is consistently dysregulated in kidney disease, liver disease, neurologic disease, collagen vascular diseases, and malignancies. On a high-fat diet, two of the most significantly downregulated microRNA groups included miR-200b and miR-200c [51].

Similarly, exercise can influence microRNA expression independent of weight loss [52]. A meta-analysis of levels of circulating microRNA in obese and lean subjects showed that physical activity had an impact on the levels of miRNAs such as miR-21, miR-126, miR-192, miR-193b, and miR-221 in diabetic and prediabetic patients [53]. The level of miR-222, which was decreased in prediabetic compared to healthy patients, was increased by physical activity in a study of healthy subjects [53]. These data suggest that circulating miRNA signatures could also monitor responses to interventions.

The type of exercise can also influence microRNA expression. Resistance training activates signaling cascades and induces epigenetic changes in pathways associated with energy metabolism and insulin sensitivity [54,55]. Endurance exercise also causes modifications in the expression of specific microRNAs that promote protein synthesis [54,55] and anti-inflammation pathways, which could be of importance in CKD.

## 4. Technical Aspects of Urinary Exosome Isolation

MicroRNA research for kidney disease [56] has suffered from technical issues that have prevented their widespread usage as biomarkers. Aside from costs, the source of the microRNA sample is important. While tissue expression or circulating serum microRNA levels remain the most used tool, urinary microRNA expression [57] is now also a valid tool with the discovery of kidney-specific microRNAs and a stable methodology to process them within exosomes in the urine.

Investigators should be diligent when analyzing the microRNA content of urinary exosomes, as urinary exosomal preparations are notorious for containing impurities that may include nonexosomal microRNA [58]. Indeed, isolating urinary exosomes is challenging due to the unique composition of urine, which contains numerous substances and materials that can contaminate exosomal preparations [59]. Therefore, obtaining highly pure urinary exosomal preparations that also contain high yields of exosomes can become tedious and complicated [60]. It is generally recognized that when isolating exosomes from biological fluids, the methods used to obtain higher protein yields from exosomal preparations often result in sacrificing the purity of these exosomal preparations [61], and specifically isolating exosomes from urine appears to be no exception [62].

Furthermore, there are other obstacles scientists may encounter when attempting to isolate urinary exosomes. For instance, the gold standard for isolating exosomes from biological fluids, including urine, is ultracentrifugation [63,64]. If laboratories do not have reliable access to an ultracentrifuge, they may need to rely on commercially available exosome isolation kits, which can be costly and possibly inapplicable when isolating exosomes from urine, and may result in urinary exosomal preparations that have suboptimal yield/purity. It should be noted that, while ultracentrifugation is indeed the gold standard for isolating exosomes from urine, it is generally acknowledged that this method still does not provide exosomal preparations devoid of contamination [58]. Incorporating gradient-based isolation techniques into urinary exosome isolation protocols, which involve ultracentrifugation, can robustly purify exosomal preparations [65]. However, using gradient-based ultracentrifugation for exosome isolation is much more labor-intensive and cumbersome [66], so difficulties may arise when laboratories attempt to utilize this technique for urinary exosome isolation.

Other techniques may also be used to isolate exosomes from urine, which include ultrafiltration, precipitation, immunoaffinity-based isolation, microfluidics, and size-exclusion chromatography [67]. However, these alternative methods do have disadvantages, including cost, time, and labor. Moreover, these methods are not as routinely used for urinary exosome isolation when compared to ultracentrifugation, so protocol optimization may prove difficult [68]. Regardless of which exosome isolation techniques are utilized, the rigorous characterization of exosomal preparations [69,70] should be used to ensure that urinary exosomes have been effectively isolated. Furthermore, depending on how urinary exosomes are to be used downstream largely impacts whether characterized exosomes are well-suited for precise downstream applications. For instance, there are various methods for enriching exosomes with transgenic microRNA [70,71], so that these modified exosomes can be used to treat cultured cells. To determine whether urinary exosomes contain a particular microRNA, an exosome degradation assay can be used [70,72,73]. However, if exosomal urinary microRNA is being used to assess clinical biomarkers via downstream omics, then exosome preparations that lack nonexosomal microRNA should only be used [74].

## 5. Future Implications

The early detection of CKD risk in all populations is key to reducing the burden of disease. There is not one clear candidate that meets all requirements for a biomarker. The answer lies in developing a matrix of clinical and biochemical markers, that, when scored together, would place each individual at a specific level of risk for the progression of CKD. Clinical calculators (e.g., https://kidneyfailurerisk.com/, accessed on 23 August 2022) exist for adults, but lack sufficient early biomarkers to increase the accuracy of prediction. This kind of scoring system would then become actionable as more therapies emerge in the future to maintain kidney function. At the very least, such a system could separate patients into low/high-risk categories, thus, enabling resource allocation to those at highest risk of developing CKD.

Moreover, it appears that different isolation techniques are capable of recovering distinct subtypes of exosomes, so prudence should be demonstrated when using exosomal microRNAs for identifying diagnostic biomarkers [74]. Fortunately, there are many kits commercially available that make microRNA extraction from exosomes both simple and straightforward, so that high-integrity exosomal microRNA may be used effectively for downstream analyses.

## Figures and Tables

**Figure 1 jcm-11-05271-f001:**
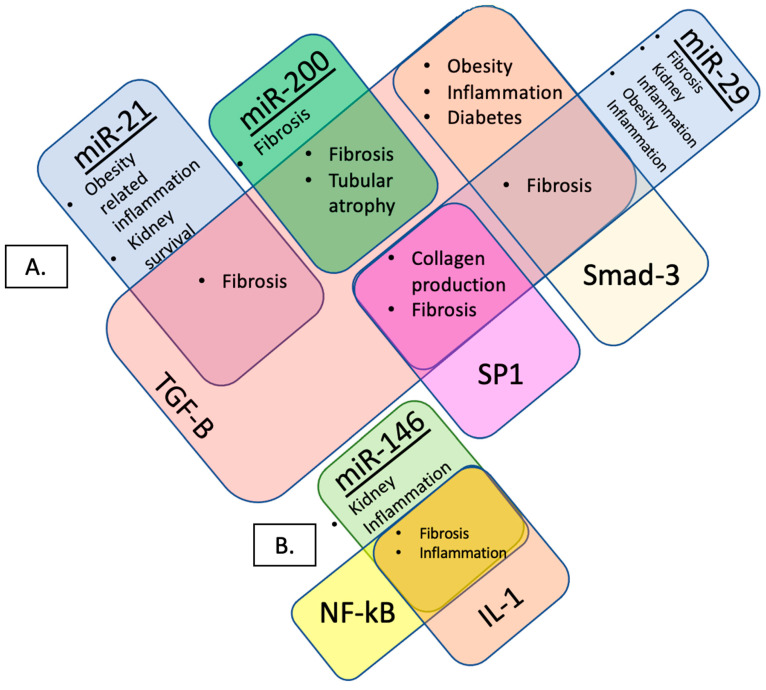
Candidate microRNAs and pathways associated with obesity, inflammation, and kidney-specific outcomes. Figure 1 shows the overlap of candidate microRNAs and pathways associated with obesity, inflammation, and kidney-specific outcomes. (**A**). miR-21 is associated with kidney fibrosis and obesity-related inflammation (via TGF-B/Smad3) and diabetes. miR-29 is associated with kidney fibrosis and inflammation (via TGF-B/Smad3 and SP1) and obesity. miR-200 is associated with kidney fibrosis and tubular atrophy (via TGF-B/SP1). (**B**). miR-146 is associated with kidney fibrosis and heightened inflammatory states (overweight and obesity (via NF-KB and IL-1)).

## Data Availability

Not applicable.

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
