# Peer review of "Urinary Exosomal MicroRNAs as Biomarkers for Obesity-Associated Chronic Kidney Disease"

_jcm, 2022, doi:10.3390/jcm11185271_

Round 1

Reviewer 1 Report

Congratulations for your work!

I just want to recommend you to uniformly use the abbreviation " miR " and not "mir"in your review. The same consideration for Figure nr 1, where you are using both MIR and Mir...

Maybe it would be useful to add some words about miR-126 and miR-770 family, considerinf the frequent association of Obesity with DM and Diabetic Nephropathy; the same recommendation for miR-103a-3p from Hypertensive Nephroangiosclerosis.

I recommend you the following article: Coonor KL, Denby L. Micro-RNAs as non-invasive biomarker in renal diseases. NDT, March 2021, 36 (3): 428-429. 

Reviewer 2 Report

The manuscript reviewed the role of microRNAs in obesity and CKD, suggesting that urinary exosomal microRNA might serve as biomarkers for obesity and CKD. The topic is of importance.

I would have some comments below.

The article’s structure includes an abstract, introduction, methods, results, discussion, and references, which seems to be a research article rather than a review article. I would recommend having a structure of abstract, introduction, body, and references. The body presents the current knowledge and understanding of a specific topic with headings and subheadings to find out recent developments in the topic, as well as what and where the gap is.

The title seems not to match the context. The body provides the role of microRNAs in kidney fibrosis, obesity, diet, and exercise. However, the title appears to focus on obesity and CKD. Please clarify them so that the audience easily follow. For example, Lines 164-165, “On a high-fat diet, two of the most significantly downregulated microRNA groups included miR-200b and miR-200c.” How does this have some impact on obesity and CKD?

Lines 174-176, “Resistance training activated signaling cascades and induced epigenetic changes in pathways associated with energy metabolism and insulin sensitivity (source?).” Please clarify “(source?)” How related to the microRNAs in obesity and CKD? 

Lines 176-178, “Endurance exercise also caused modifications in the expression of specific miRNAs that promote protein synthesis [43, 177,44] and anti-inflammation pathways which could be of importance in CKD.” Please restate it in detail regarding urinary exosomal miRNAs in CKD, taken together in Figure 1.

Concerning the urinary microRNA, it would be helpful for the audience to follow to state the issues and possible approaches with the particular headings and subheadings.

Future perspectives should be stated in the last paragraph instead of in the middle.

Reviewer 3 Report

The type of the article is not clear

It is a review article - Materials and Methods are not clear 

There is no method specified in this section. Is it a Meta-analysis? What database was searched? What timeframe was used to search the data?

For the review article  -this section is not required, instead, just describes the different markers and the Current scope of CKD biomarker research

Reference parenthesis was not used correctly, [   ] used after the period in almost every place eg. line 61,72,81.........  It should be used before the period with a space, eg;  sentence ....... [   ].

In line 138, remove PubMed ID

The result section is not clear. Not sure what result was mentioned, just a description of the process and pathogenesis- should highlight the title: Effect of diet and exercise on MicroRNA expression

In a review article, the result is not necessary

The last paragraph of the result section - there is no ref, it needs a ref

Line 176- needs ref

1st Paragraph of the Discussion section needs more ref

Line 191- needs ref

Needs minor grammatical revision 

Round 2

Reviewer 3 Report

Changes are appropriate